# Ester Linked Fatty Acid (ELFA) method should be used with caution for interpretating soil microbial communities and their relationships with environmental variables in forest soils

**Wenjuan Yu[1,2], Huanhuan Gao[3], Hongzhang Kang[1]\***

1 Department of Landscape Architecture, School of Design, Shanghai Jiao Tong University, Shanghai, China, 2 Department of Ecology, Evolution, and Organismal Biology, Iowa State University, Ames, IA, United States of America, 3 School of Agriculture and Biology, Shanghai Jiao Tong University, Shanghai, China

\* kanghz@sjtu.edu.cn

## Abstract

As an alternative for phospholipid fatty acid (PLFA) analysis, a simpler ester linked fatty acid (ELFA) analysis has been developed to characterize soil microbial communities. However, few studies have compared the two methods in forest soils where the contribution of nonmicrobial sources may be larger than that of microbial sources. Moreover, it remains unclear whether the two methods yield similar relationships of microbial biomass and composition with environmental variables. Here, we compared PLFA and ELFA methods with respect to microbial biomass and composition and their relationships with environmental variables in six oriental oak (*Quercus variabilis*) forest sites along a 1500-km latitudinal gradient in East China. We found that both methods had a low sample-to-sample variability and successfully separated overall community composition of sites. However, total, bacterial, and fungal biomass, the fungal-to-bacterial ratio, and the gram-positive to gram-negative bacteria ratio were not significantly or strongly correlated between the two methods. The relationships of these microbial properties with environmental variables (pH, precipitation, and clay) greatly differed between the two methods. Our study indicates that despite its simplicity, the ELFA method may not be as feasible as the PLFA method for investigating microbial biomass and composition and for identifying their dominant environmental drivers, at least in forest soils.

## Introduction

One widely used approach for characterizing microbial communities in environmental samples is phospholipid fatty acid (PLFA) analysis. Since microbial PLFAs from cell membranes are rapidly degraded upon cell death, they can be considered to be representative of the viable microbes [1]. PLFA analysis can be applied to measure microbial biomass and composition [2]. In this multiple-step method, microbial lipids are usually extracted from environmental samples in a phase mixture of chloroform, methanol, and buffer. Lipids associated with the organic phase are then fractionated into neutral, glyco-, and phospholipids on silicic acid

**Data Availability Statement:** All relevant data are within the manuscript and its Supporting Information files.

**Funding:** This work was supported by National Natural Science Foundation of China (31770746, 31270491). The funders had no role in study design, data collection and analysis, decision to publish, or preparation of the manuscript.

**Competing interests:** The authors have declared that no competing interests exist.

columns. Finally, the phospholipids are subjected to mild alkaline methylation to produce fatty acid methyl esters (FAMEs) for gas chromatography analysis.

Considering that the PLFA method is arduous and time-consuming, a simpler base-catalyzed method for extraction and transesterification of soil fatty acid esters, i.e. ester linked fatty acid (ELFA), has been developed [3]. Although less popular than PLFA, this technique has been extensively used to investigate microbial communities in various ecosystems including forests [4], prairies [5] and agricultural fields [6]. Without lipid extraction and separation, the ELFA procedure uses a mild alkaline reagent to lyse cells and release fatty acids as methyl esters from soil lipids in one step [3]. Unlike PLFAs that only come from viable cell membranes, the ELFA method is non-specific: fatty acids from phospholipids, glycolipids and neutral lipids are extracted from intact microbial cells as well as from dead organic material [7]. Thus, despite the fact that the ELFA method is faster, simpler and more sensitive than the PLFA method, it is more difficult to draw conclusions about changes in the extant microbial community based on the ELFA results and there is a strong need to estimate ELFA vs. PLFA.

Studies have compared the two methods for characterizing soil microbial communities in native sod and wheat-fallow plots [8], during different stages of composting process [7], in soils with varying degrees of metal pollution [9], in forest soils with different tree types and nitrogen loads [4], and in forest and arable soils [10]. Although the two methods sometimes did not yield consistent changes in the composition of fatty acids or microbial groups among the soils, these studies all showed that both methods were able to separate overall community composition of different soils. However, none of the studies have fully investigated if the two methods yielded consistent relationships of microbial biomass and composition with environmental drivers. These relationships are important for predicting the responses of microbial communities and their controls over biogeochemical processes under the context of accelerating global change [11].

Here, we investigated soil microbial communities using the PLFA and ELFA methods from six oriental oak (*Quercus variabilis*) forest sites arranged across a 11$^\circ$ latitudinal gradient in East China. Covering a wide range of climatic conditions and soil physicochemical properties (S1 Table), these sites offered a unique opportunity to estimate if the two methods yield consistent results of how environmental variables drive changes in soil microbial communities. The goals of the study were to compare the two methods with respect to their efficiency for discriminating among different sites with regard to 1) overall community composition, 2) microbial biomass and specific groups, and 3) relationships of soil and climatic variables with overall community composition, microbial biomass and specific groups.

## Materials and methods

### Soil sample collection

The sampling procedure has been described in detail by Kang et al. [11]. In July 2016, six oriental oak sites in East China were sampled along a 1500-km climatic gradient, i.e., PG, HYS, BA, HZY, XY and YS (S1 Table). At each site, on each of the three randomly chosen transects, mineral soils at 0–10 cm depth from 10 sampling points (at 5 m spacing) were collected with a 2.5-cm corer and composited, resulting in a total of 18 independent soil samples (6 sites × 3 replicates). Soil samples were placed in polyethylene bags, stored on ice and transferred to the laboratory within 24 h. Soils were stored at 4˚C for less than one week, with early-collected soils stored longer and later-collected soils stored shorter. Soils were sieved (2mm) with any remaining visible plant material and stone removed by hand. Each sample was divided into two subsamples, one for soil physicochemical analysis, and the other for PLFA and ELFA analyses.

## Soil physicochemical analysis

Fresh soils were air-dried, pulverized and screened through mesh size of 80 (0.18 mm) for soil physicochemical analysis. Soil C (SOC) and N contents were determined by a Vario EL III elemental analyzer (Elementar Analysensysteme GmbH, Germany). Soil pH was measured at a soil-to-water ratio of 1:2.5 (m/v). Soil particles were classified into silt, clay, and sand by the Bouyoucos hydrometer method [12]. Soil moisture was determined by comparing the weights of soils before and after oven-drying at 105°C.

## PLFA extraction

As shown in the left side of Fig 1, lipids were extracted from 10 g of fresh soil using a single-phase mixture of chloroform, methanol, and citrate buffer (pH 4.0) in a ratio of 1:2:0.8 [13–15]. Lipid classes were then fractionated into neutral, glyco- and phospholipids using solid phase extraction chromatography. Fatty acids in the phospholipid were converted to FAMEs by a mild alkaline methylation and extracted by hexane. The FAMEs were dried at room temperature under nitrogen stream and redissolved in 200 μL of hexane for instrumental analysis. Methyl nonadecanoate (C19:0 FAME) was used as internal standard.

## ELFA extraction

A detailed description of ELFA procedures are available in Schutter and Dick [3]. The reagent volumes were modified according to soil weight. As shown by the right side of Fig 1, 10 g of fresh soil was added with 0.2 M KOH in methanol and incubated at 37°C for 1h with moderate vortexing, during which ELFAs were released and methylated. 1.0 M acetic acid was added to neutralize the pH. Like above, FAMEs were partitioned into hexane, dried under nitrogen stream, and dissolved in 1 mL of hexane: methyl-*tert* butyl ether (1:1, v/v) for instrumental analysis. C19:0 FAME was also used as internal standard.

## Gas Chromatography Mass Spectrometry (GC-MS) analysis

FAME analysis was performed on a 7890A gas chromatograph coupled with a 5975C mass spectrometer (Agilent, USA). The MSD ChemStation software was used to analyze data. FAME peaks were identified by comparing their retention times and mass spectra with those from reference compounds and from the NIST 2014 spectral database. Fatty acid concentrations were calculated according to peak areas using the internal standard method and reported as nmol $g^{-1}$ dry soil.

43 fatty acids were included in total biomass and used for overall community composition analysis. They were common to both methods. Saprotrophic and ectomycorrhizal fungal (hereafter fungal) biomass was represented by 18:2ω6,9 and 18:1ω9 [16, 17]. Nine fatty acids were summed to represent bacterial biomass: i15:0, a15:0, i16:0, i17:0, a17:0, 16:1ω9, 16:1ω7, cy17:0 and cy19:0, with the first five and the last four used as biomarkers for gram-positive (G+) and gram-negative (G-) bacteria, respectively [15, 18, 19]. The fatty acid C16:1ω5 was used as the biomarker for arbuscular mycorrhizal fungi (AMF) [17]. The fungal-to-bacterial (F/B) ratio was calculated by dividing the sum of two fungal biomarkers through all the bacterial biomarkers. The ratio of gram-positive to gram-negative bacteria (G+/G-) was calculated by dividing the gram-positive sum by the gram-negative sum. Relative abundance of a specific group or fatty acid was calculated by dividing its biomass by total biomass.

## Statistical analysis

Site-level coefficient of variation (CV) for each of the 43 fatty acids was calculated and averaged across all fatty acids to compare method precision. Redundancy analysis (RDA) based on

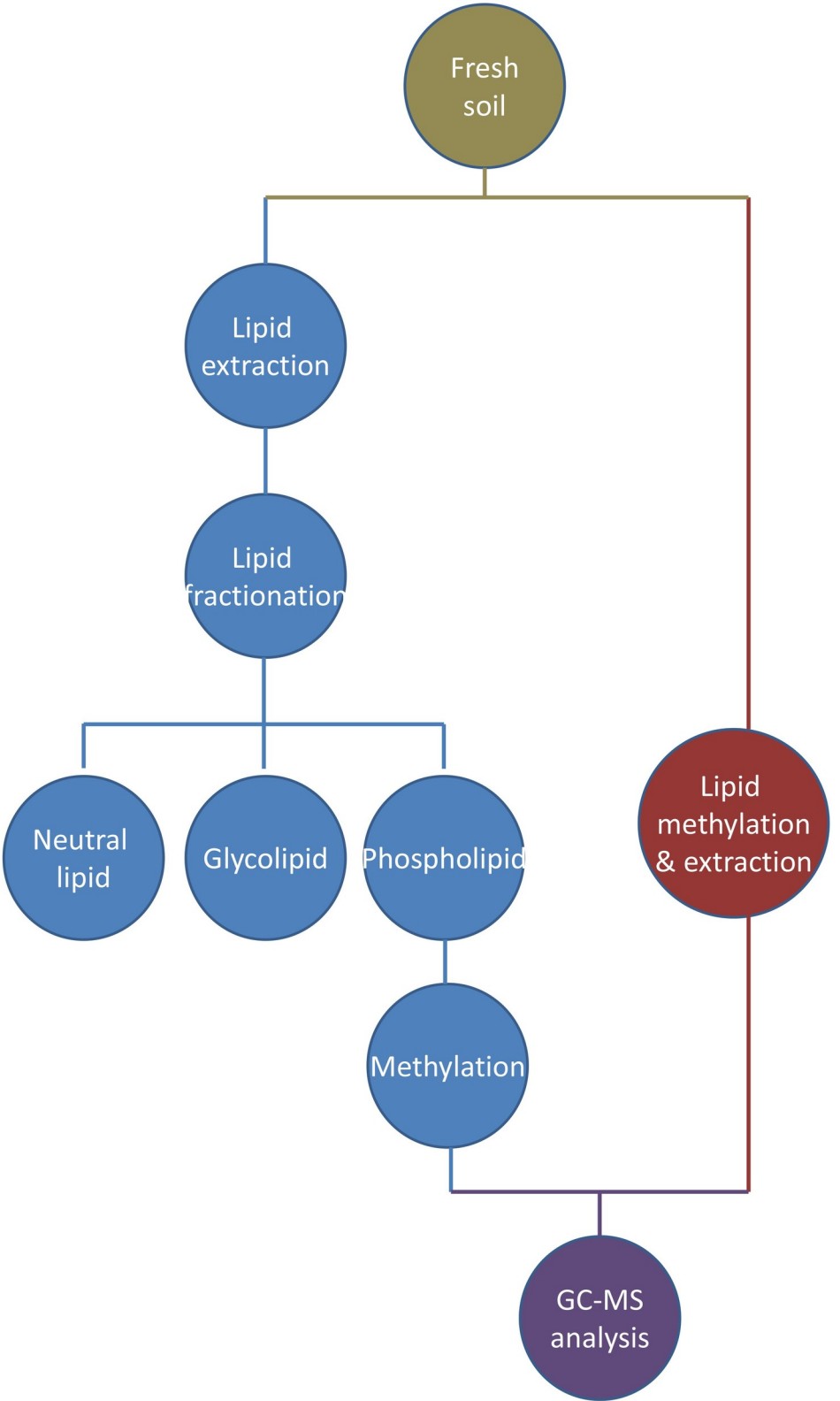

**Fig 1. Protocol workflow for PLFA (left) and ELFA (right) methods.** GC-MS, gas chromatography-mass spectrometry.

relative abundances of all fatty acids was used to perform constrained ordination and to summarize relationships among overall community composition and environmental variables in Canoco for Windows 4.5 (Biometris, Netherlands). Seven variables possibly related to microbial composition were included, i.e., pH, sand, clay, SOC, C/N ratio, mean annual precipitation (MAP), and altitude. Silt, total N, and mean annual temperature (MAT) were excluded due to their collinearities with one or more of the seven variables. Variable significance ($P < 0.05$) was assessed by Monte Carlo permutation tests. Pearson correlations were used to examine bivariate relationships between PLFA- and ELFA-derived microbial properties and between microbial properties and the seven environmental variables in the SPSS 19.0 software (SPSS Inc., USA). Six microbial properties important for microbial communities were selected: total, bacterial, and fungal biomass, F/B ratio, G+/G- ratio, and relative abundance of AMF. The number of samples (n = 18) may be relatively small compared with the number of independent variables (environmental variables, n = 7) for each dependent variable (microbial variable), yet we focused on bivariate correlations instead of identifying environmental variables that can best predict microbial properties. Bonferroni adjustment of P value was used for correlation analyses to correct for multiple comparisons and the threshold for adjusted P values was 0.10.

## Results and discussion

Both the multiple-step PLFA and the one-step ELFA methods showed a low sample-to-sample variability. Site-level coefficient of variation (CV) averaged across absolute abundances of 43 fatty acids ranged between 5.0% and 14.1% for PLFA and between 5.2% and 15.6% for ELFA, respectively (Table 1). Drenovsky et al. [20] reported a similar precision of the PLFA method, with mean CV ranging from 4.8% to 17.9%. However, they found a much higher CV (18.1%– 66.4%) for the MIDI method, possibly due to the relatively drastic extraction conditions. As another popular alternative of PLFA, the MIDI protocol targets soil lipids similar to those in the ELFA protocol, which are saponified by heat and the addition of a strong base and methylated through acid catalysis. Previous studies comparing PLFA and ELFA did not report results of overall method precision [9, 10]. Our study suggests that both the PLFA and ELFA methods have a relatively high precision for quantifying microbial fatty acids.

The RDA plots showed that both methods were able to clearly differentiate among sites (Fig 2). Overall community composition based on PLFAs showed a similar distribution pattern to the composition based on ELFAs. Three soils from each site grouped tightly, consistent with the low sample-to-sample variability within each site shown by both methods (Table 1). The two southern sites (XY and YS) were distinct from the four more northerly sites (PG, HYS, BA, and HZY) and from each other. The first two axes explained a high proportion (79.1%) of the sample variation based on PLFA composition; they explained a similarly high proportion (80.8%) of the variation based on ELFA composition. Our results, together with previous studies [8, 9], show that both methods could successfully separate different soils. This indicates that

**Table 1. Mean coefficient of variation across 43 fatty acids for each site for PLFA and ELFA methods.**

| Site | PLFA (%) | ELFA (%) |
|------|----------|----------|
| PG | 14.1 | 9.7 |
| HYS | 12.5 | 7.6 |
| BA | 5.0 | 15.6 |
| HZY | 9.0 | 5.2 |
| XY | 11.3 | 13.9 |
| YS | 10.1 | 9.2 |

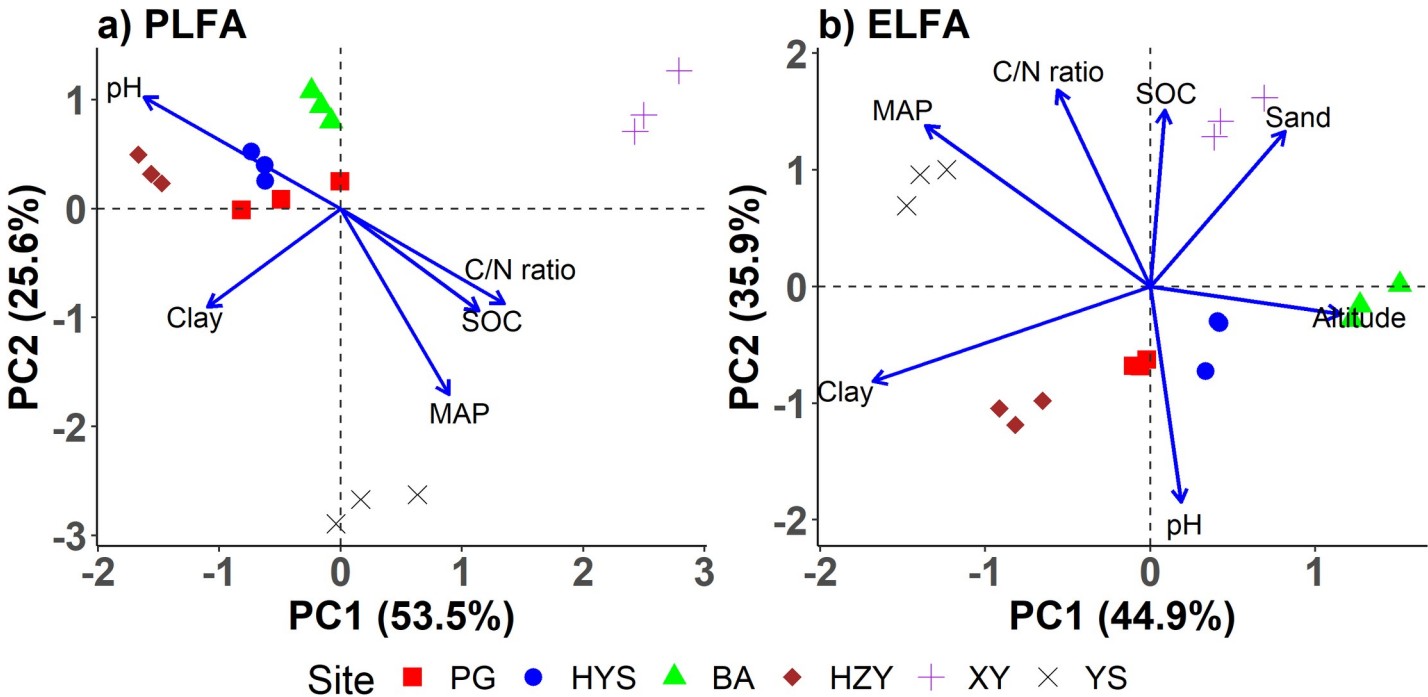

**Fig 2.** a) Redundancy analysis of soil microbial community composition using PLFAs across six oriental oak sites in East China. Two insignificant (P > 0.05) variables, i.e., sand and altitude, are removed. b) Redundancy analysis of soil microbial community composition using ELFAs.

the non-specific ELFA method may give reasonable estimates of overall microbial community composition.

However, the two methods did not yield consistent results regarding absolute and relative abundances of specific groups. Like previous studies [4, 10], absolute abundances of ELFAs were generally higher than those of PLFAs (S2 and S3 Tables); mean total ELFAs across all samples (317 nmol g$^{-1}$) was roughly threefold higher than mean total PLFAs (96 nmol g$^{-1}$). Moreover, ELFA-derived groups were generally not related to PLFA-derived groups (Fig 3). Total, bacterial, and fungal PLFAs were insignificantly (P > 0.05) correlated with total, bacterial, and fungal ELFAs, respectively. F/B ratio based on PLFAs was insignificantly correlated with that based on ELFAs. G+/G- ratio based on PLFAs showed a significant but weak relationship with that based on ELFAs (r = 0.49, P < 0.05). Relative abundance of AMF was relatively consistent between the two methods (r = 0.66, P < 0.05). These results are inconsistent with a previous study reporting much stronger positive relationships (r$^2$ ranged from 0.96 to 1) between the methods with respect to fungal and bacterial biomass, relative abundance of fungi, and F/B ratio in forest soils [10]. In two non-forest studies, some fatty acids or microbial groups showed consistent responses to metal pollution [9] and composting time [7] between the two methods, while some showed inconsistent responses.

Ruling out the effect of soil storage, the inconsistency in these microbial parameters between methods indicate that fatty acids other than microbial phospholipids were also extracted by ELFAs. For example, *iso-* and *anteiso*-branched fatty acids have been extracted from bacterial membranes as well as humin fractions [21]; a biomarker for fungi (18:2ω6,9) is also a constituent of plant membranes. The consistency in AMF abundance may be a coincidence because the biomarker for AMF (16:1ω5) can derive from mixed sources in both methods. 16:1ω5 extracted by ELFA might come more from neutral lipids than from phospholipids [22], while 16:1ω5 obtained by PLFA could be contributed by both AMF and some bacteria

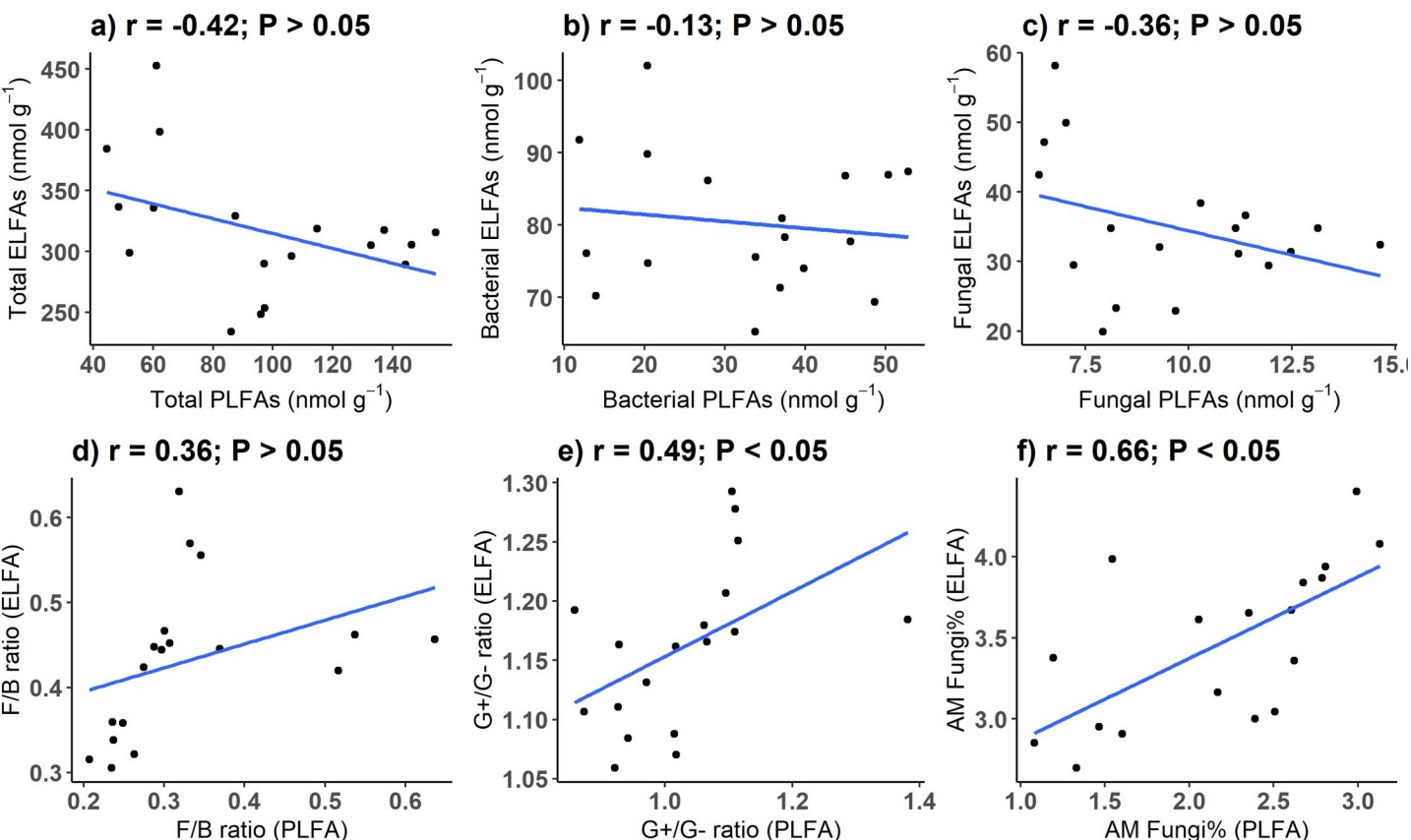

**Fig 3. Bivariate correlations between PLFA and ELFA biomarkers.** a) total PLFAs vs. total ELFAs; b) bacterial PLFAs vs. bacterial ELFAs; c) fungal PLFAs vs. fungal ELFAs; d) the ratio of fungi to bacteria (PLFA) vs. the ratio of fungi to bacteria (ELFA); e) the ratio of gram-positive bacteria to gram-negative bacteria (PLFA) vs. the ratio of gram-positive bacteria to gram-negative bacteria (ELFA); f) relative abundance of arbuscular mycorrhizal fungi (PLFA) vs. relative abundance of arbuscular mycorrhizal fungi (ELFA).

including *Cytophaga* and *Flexibacter* [23]. Overall, our results indicate that microbial biomass and specific groups, as important indicators of soil microbial communities [24, 25], were dependent on the extraction methods. Care should be taken when using ELFA to estimate microbial biomass and composition in soils where living organisms may not account for the majority of extracted fatty acids.

The relationships of microbial biomass and composition with environmental drivers were also inconsistent between the two methods. In the RDA plots (Fig 2), five variables were significantly ($P < 0.05$) linked with the first two axes of overall PLFA composition, i.e., MAP ($r^2 = 0.93$), pH ($r^2 = 0.91$), C/N ratio ($r^2 = 0.64$), SOC ($r^2 = 0.54$), and clay ($r^2 = 0.50$); all seven variables were significantly associated with the first two axes of overall ELFA composition, i.e., MAP ($r^2 = 0.94$), clay ($r^2 = 0.87$), pH ($r^2 = 0.86$), C/N ratio ($r^2 = 0.79$), sand ($r^2 = 0.61$), SOC ($r^2 = 0.57$), and altitude ($r^2 = 0.35$). More specifically (Fig 4), total PLFAs was significantly (adjusted $P < 0.1$) correlated with pH, sand, and clay and bacterial PLFAs was significantly correlated with clay, but total and bacterial ELFAs were significantly correlated with none of the environmental variables. The relationships among fungal biomass and F/B ratio and environmental variables also differed with extraction method. Only relative abundance of AMF had positive relationships with pH for both methods, consistent with a study investigating microbial responses to heavy metal-polluted soils [9]. However, Hinojosa et al. [9] also reported that pH and metals correlated with monounsaturated to saturated fatty acid ratio

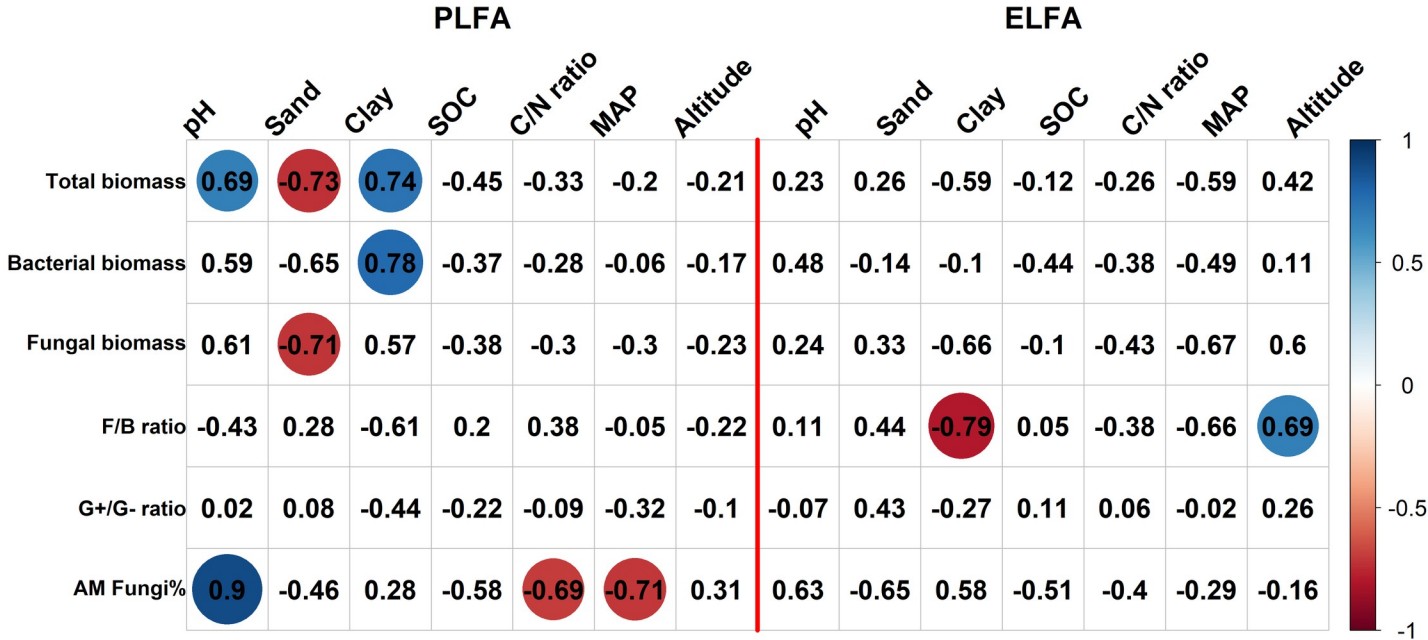

**Fig 4. Bivariate correlations among microbial and environmental variables.** The left seven columns are determined based on PLFA biomarkers and the right seven columns are determined based on ELFA biomarkers. The color indicates the direction of correlation and the number represents the correlation coefficient. A circle is present if Bonferroni-adjusted *P* < 0.1, with circle size indicating strength of the correlation. F/B ratio, the ratio of fungi to bacteria; G+/G− ratio, the ratio of gram-positive bacteria to gram-negative bacteria; AM Fungi%, relative abundance of arbuscular mycorrhizal fungi; MAP, mean annual precipitation.

based on PLFAs but on ELFAs. We would not interpret the relationships of PLFA composition with environmental variables, which could be found in Kang et al. [11]. Yet our findings indicate that based on ELFAs, we might draw totally different conclusions about how the surroundings have shaped microbial communities and how future environmental changes would possibly drive changes in microbial communities, as compared with those based on PLFAs. This is risky given that ELFAs comprise both microbial and nonmicrobial fatty acids. The different relationships between environmental variables and microbial communities indicate weak correlations between the two methods.

In conclusion, both the PLFA and ELFA methods showed a high method precision and successfully separated overall soil microbial community composition of six forest sites. Given the simplicity of the ELFA method, it may be a useful tool for routine soil monitoring, such as discrimination among different treatments, sites or stages. However, microbial biomass and important microbial groups as well as their relationships with environmental variables differed greatly between the two methods. We suggest that the ELFA method should be used with great caution for interpreting changes in microbial biomass and composition and for identifying dominant drivers of these changes, at least in forest soils.

## Supporting information

**S1 Table. Climatic and soil characteristics at the six oriental oak (*Quercus variabilis*) forest sites in eastern China.**
(DOCX)

**S2 Table. Site-averaged absolute abundances of fatty acids (nmol g⁻¹) for PLFA and ELFA methods.**
(DOCX)

**S3 Table. Site-averaged relative abundances of fatty acids (%) for PLFA and ELFA methods.**
(DOCX)

**S1 Data.**
(XLSX)

## Author Contributions

**Investigation:** Wenjuan Yu, Huanhuan Gao, Hongzhang Kang.

**Methodology:** Wenjuan Yu, Hongzhang Kang.

**Writing – original draft:** Wenjuan Yu, Hongzhang Kang.

**Writing – review & editing:** Hongzhang Kang.

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
