## [Decision Letter · Decision Letter 0]

22 Mar 2021

PONE-D-20-33719

Ester linked fatty acid (ELFA) method should be used with caution for interpretating soil microbial communities and their relationships with environmental variables in forest soils

PLOS ONE

Dear Dr. Kang,

Thank you for submitting your manuscript to PLOS ONE. After careful consideration, we feel that it has merit but does not fully meet PLOS ONE’s publication criteria as it currently stands. Therefore, we invite you to submit a revised version of the manuscript that addresses the points raised during the review process. In particular, please address the concerns of reviewer #2.  To be conclusive, further statistical analyses are needed and the sample storage conditions must be specified. In the absence of these important revisions, I ask that you please provide a compelling counter argument. Beyond these significant points, both reviewers offer relatively minor suggestions that will improve the paper.

We look forward to receiving your revised manuscript.

Kind regards,

Daniel Cullen

Academic Editor

PLOS ONE

Journal Requirements:

Reviewers' comments:

Reviewer's Responses to Questions

**Comments to the Author**

1. Is the manuscript technically sound, and do the data support the conclusions?

Reviewer #1: Yes

Reviewer #2: No

2. Has the statistical analysis been performed appropriately and rigorously? 

Reviewer #1: Yes

Reviewer #2: No

3. Have the authors made all data underlying the findings in their manuscript fully available?

Reviewer #1: Yes

Reviewer #2: Yes

4. Is the manuscript presented in an intelligible fashion and written in standard English?

Reviewer #1: Yes

Reviewer #2: Yes

5. Review Comments to the Author

Reviewer #1: Yu and colleagues compared the efficiency of two methods (phospholipid fatty acid (PLFA) analysis and ester linked fatty acid (ELFA) analysis), in the characterization of forest soil microbial communities. The topic is interesting, because it is really good to know the efficiency of the different methods in order to be able to choose the most suitable one to find the best response to the biological questions.

General comments:

In my opinion the article is well written, the results are presented in a clear way and the conclusion are supported by the data. I have only some minor comments:

Lines 167-170: the authors found that the coefficient of variation among the different sites is in most cases higher for the PLFA technique, except in the case of sites BA and XY. Can they comment a bit this point? Which is the difference among BA and XY sites and the others analyzed? The answer to this question will give some more information about the different efficiency of the two methods.

Lines 172-175: can the authors comment about the fact that the MIDI method produce a so high CV compared to ELFA?

Lines 211-213: the authors found that ELFA and PLFA gave similar results for the relative abundance of AMF. It’s curious that the two methods arrive to similar results only in the case of this particular soil microbes. Can the authors try to comment this result?

Reviewer #2: Overview: Overall this study is straightforward, simple to understand, and the manuscript is well-written. There are some important changes that should be made before I can recommend this paper for publication. Importantly, information on how the soil was stored in the lab needs to be provided to illustrate that the weak correlations between the PLFA and ELFA data was not due to improper storage. Secondly, additional analysis should be done to support the argument that high organic matter influenced ELFA results. Finally, additional univariate analyses should be performed to account for all environmental variables. Specific comments follow.

Introduction:

Line 62: I suggest revising to “Studies have compared the two methods for characterizing soil microbial communities in native sod and wheat-fallow plots [9], during different stages of composting [7], in soils with varying degrees of metal pollution [9], in forest soils with different tree types and nitrogen loads [4], and in forest and arable soils [10].

Lines 65-67: I suggest revising to “Although the two methods sometimes did not yield consistent results in the composition of fatty acids or microbial groups among the soils…”

Line 77: Revise “physic-chemical” to “physicochemical”. Make this change throughout the manuscript.

Materials and Methods:

In the sampling section, the authors state that samples were transferred to the laboratory within 24 h. However, they do not specify how they were stored in the lab (i.e what temperature), nor for how long they were stored. The correlations between the two methods presented here are substantially weaker than other studies that have compared these methods, including for studies utilizing forest soils. One reason for this weak correlation may be improper soil storage. In order to alleviate this concern, the authors should provide more information about how the soils were stored in the lab.

Line 97: Revise “smashed” to “pulverized”.

Line 102: Revise “decided” to “determined”.

Line 118: I suggest revising to “A detailed description of ELFA extraction procedures are available in Schutter and Dick [3]. The reagent volumes were modified according to soil weight.”

Line 134-135: Although many publications refer to these two lipids as saprotrophic fungal indicators, these lipids are also abundant in ectomycorrhizal fungi (see https://doi.org/10.1111/j.1574-6941.1999.tb00621.x). Therefore, it is incorrect to refer to “saprotrophic fungal biomass” here and elsewhere. Revise to “fungal biomass” or “saprotophic + ectomycorrhizal biomass.”

Line 146: I don’t think it is necessary to define coefficient of variation for a science audience.

Line 154: Total N was excluded due to collinearity, presumably because it is collinear with C/N ratio. Why, then, was SOC included when it is likely also collinear with C/N? If variables need to be excluded due to collinearity, it would be better to exclude C/N and include N. Relationships between SOC and N separately tend to be more informative than C/N because they identify the specific driver of responses.

Lines 161-163: The authors selected only three environmental variables for analysis in part because there were only 18 samples. However, the authors performed univariate correlations, so 18 samples is not restrictive to analyze all seven environmental variables. The relationships between environmental variables and the lipids is the most novel part of this study. I suggest that the authors analyze univariate relationships of all possible variables.

Results and Discussion:

Currently, I find the authors claims that high organic matter accounts for the lack of correspondence between the two methods to be unconvincing for several reasons. First, the SOC in their soils ranged from 2.1-3.8%. This is not unusually high and is within the range of other studies that have found good agreement between the two methods. Additionally, the two techniques were previously compared in forest soils in a study that found much stronger correlations than in the current study. Agreement was also found between the methods in compost, which was presumably very high in organic matter. The authors could support this argument if they showed that ELFA was more strongly correlated with SOC than was PLFA, but they have not performed this analysis.

The most interesting part of this study is the relationships between the fatty acids and environmental variables, but, as stated above, the authors didn’t analyze this data extensively, which has left this portion of the paper quite weak. The authors should perform additional univariate analyses between environmental variables and the fatty acid groupings. The relationships between environmental variables and fatty acids could also be invoked to be explain the weak correlations between the ELFA and PLFAs.

Figure 2a: “SOC” and “CN Ratio” are overlapping in this figure, making it difficult to read. Modify the figure so these words are not overlapping.

6. PLOS authors have the option to publish the peer review history of their article (what does this mean?). If published, this will include your full peer review and any attached files.

Reviewer #1: No

Reviewer #2: No

---

## [Author Response · Author response to Decision Letter 0]

12 Apr 2021

RESPONSE TO REVIEWER 1

Reviewer #1: Yu and colleagues compared the efficiency of two methods (phospholipid fatty acid (PLFA) analysis and ester linked fatty acid (ELFA) analysis), in the characterization of forest soil microbial communities. The topic is interesting, because it is really good to know the efficiency of the different methods in order to be able to choose the most suitable one to find the best response to the biological questions.

General comments:

In my opinion the article is well written, the results are presented in a clear way and the conclusion are supported by the data. I have only some minor comments:

Response: We thank the reviewers for their thoughtful and helpful consideration of our paper. 

Lines 167-170: the authors found that the coefficient of variation among the different sites is in most cases higher for the PLFA technique, except in the case of sites BA and XY. Can they comment a bit this point? Which is the difference among BA and XY sites and the others analyzed? The answer to this question will give some more information about the different efficiency of the two methods.

Response: This is an interesting observation, but we are cautious about the difference. The two methods both had relatively small CV (< 16%). We have not observed distinct differences in above vegetation between sites BA and XY and the other sites. Table S1 showed that the two sites both had smaller clay contents, but it is unclear if clay content was related to the variability of the two methods.

Lines 172-175: can the authors comment about the fact that the MIDI method produce a so high CV compared to ELFA?

Response: Drenovsky et al. (2004) have not proposed possible reasons for higher CV of MIDI methods compared to PLFA method in their paper. We speculate that the relatively drastic conditions used to extract fatty acids from lipids might lead to higher sample-to-sample variability. They used an 80oC water bath and a concentrated base catalyst (3.25 M NaOH) during the saponification step and a concentrated acid catalyst (6.0 M HCl:MeOH (1:0.85)) during methylation in the MIDI method. We used a 37oC incubation and a low-concentration base (0.2 M KOH in MeOH) during methylation in the ELFA method. We have added this speculation on line 177. 

Lines 211-213: the authors found that ELFA and PLFA gave similar results for the relative abundance of AMF. It’s curious that the two methods arrive to similar results only in the case of this particular soil microbes. Can the authors try to comment this result?

Response: Yes, it seems that the two methods were more consistent when using one marker than using several markers. But we cannot rule out the possibility of coincidence. The marker for AMF, i.e., 16:1ω5, can be attributed to multiple sources in both methods as we have mentioned on lines 228-232. We prefer to be more conservative about this consistency since we cannot find more papers to support it. For example, in Table 4 of the Drijber et al. (2000) paper, PLFA-derived C16:1ω5/C16:0 was always significantly different from ELFA-derived C16:1ω5/C16:0. 

Drijber RA, Doran JW, Parkhurst AM, Lyon DJ. Changes in soil microbial community structure with tillage under long-term wheat-fallow management. 314 Soil Biol Biochem. 2000;32: 1419–1430. doi:10.1016/S0038-0717(00)00060-2

RESPONSE TO REVIEWER 2

Reviewer #2: Overview: Overall this study is straightforward, simple to understand, and the manuscript is well-written. There are some important changes that should be made before I can recommend this paper for publication. Importantly, information on how the soil was stored in the lab needs to be provided to illustrate that the weak correlations between the PLFA and ELFA data was not due to improper storage. Secondly, additional analysis should be done to support the argument that high organic matter influenced ELFA results. Finally, additional univariate analyses should be performed to account for all environmental variables. Specific comments follow.

Response: We thank the reviewers for their thoughtful and helpful consideration of our paper. We have performed additional univariate analyses and specified sample storage conditions. Beyond these significant points, we accepted almost all of the additional specific suggestions, and these changes are detailed below.

Introduction:

Line 62: I suggest revising to “Studies have compared the two methods for characterizing soil microbial communities in native sod and wheat-fallow plots [9], during different stages of composting [7], in soils with varying degrees of metal pollution [9], in forest soils with different tree types and nitrogen loads [4], and in forest and arable soils [10].

Response: Thanks for your careful reading. We have made the change on lines 62-65.

Lines 65-67: I suggest revising to “Although the two methods sometimes did not yield consistent results in the composition of fatty acids or microbial groups among the soils…”

Response: We have made the change on lines 66-67.

Line 77: Revise “physic-chemical” to “physicochemical”. Make this change throughout the manuscript.

Response: We have made the change throughout the manuscript.

Materials and Methods:

In the sampling section, the authors state that samples were transferred to the laboratory within 24 h. However, they do not specify how they were stored in the lab (i.e what temperature), nor for how long they were stored. The correlations between the two methods presented here are substantially weaker than other studies that have compared these methods, including for studies utilizing forest soils. One reason for this weak correlation may be improper soil storage. In order to alleviate this concern, the authors should provide more information about how the soils were stored in the lab.

Response: Soils were stored at 4°C for less than one week, with early-collected soils stored longer and later-collected soils stored shorter. We immediately performed the PLFA and ELFA analyses after collecting all the soils. We have clarified soil storage conditions on lines 93-94. The weak correlation may not be due to soil storage. We have added this point on line 223.

Line 97: Revise “smashed” to “pulverized”.

Line 102: Revise “decided” to “determined”.

Line 118: I suggest revising to “A detailed description of ELFA extraction procedures are available in Schutter and Dick [3]. The reagent volumes were modified according to soil weight.”

Response: We have made the changes on lines 99, 104, and 120-121.

Line 134-135: Although many publications refer to these two lipids as saprotrophic fungal indicators, these lipids are also abundant in ectomycorrhizal f Revise to “fungal biomass” or “saprotrophic + ectomycorrhizal biomass.”ungi (see https://doi.org/10.1111/j.1574-6941.1999.tb00621.x). Therefore, it is incorrect to refer to “saprotrophic fungal biomass” here and elsewhere.

Response: Yes, we agree with you. This is a very classic paper. We have cited it here. We have changed “saprotrophic fungal biomass” to “saprotrophic and ectomycorrhizal fungal (hereafter fungal) biomass” on line 137. We have deleted “saprotrophic” throughout the manuscript. 

Line 146: I don’t think it is necessary to define coefficient of variation for a science audience.

Response: We have deleted the definition on line 149. 

Line 154: Total N was excluded due to collinearity, presumably because it is collinear with C/N ratio. Why, then, was SOC included when it is likely also collinear with C/N? If variables need to be excluded due to collinearity, it would be better to exclude C/N and include N. Relationships between SOC and N separately tend to be more informative than C/N because they identify the specific driver of responses.

Response: The reviewer raises an important point, although we respectfully disagree with the statement that “it would be better to exclude C/N and include N”. Since most (> 95%) of soil N is organic, both SOC and N concentrations are useful indicators of SOM quantity. SOC is typically correlated with N. Cleveland and Liptzin (2007) reported that on average, atomic C:N:P ratio in the soil is 186:13:1, and this ratio is well-constrained at the global scale. 

Lower soil C/N ratios usually indicate greater microbial processing, as microbes generally have lower C/N ratios (varying between 8:1 and 12:1) than those of plant litter (varying between 10:1 and 100:1). 

Indeed, we found a strong positive correlation between SOC and N (r = 0.93), which could cause a serious collinearity problem. C/N ratio had a weaker correlation with SOC (r = 0.69). 

Thus, we believe SOC and C/N ratio are two important soil properties that need to be included in the statistical analyses.

Cleveland CC, Liptzin D. C:N:P stoichiometry in soil: is there a “Redfield ratio” for the microbial biomass? Biogeochemistry. 2007;85: 235–252. doi:10.1007/s10533-007-9132-0 

Lines 161-163: The authors selected only three environmental variables for analysis in part because there were only 18 samples. However, the authors performed univariate correlations, so 18 samples is not restrictive to analyze all seven environmental variables. The relationships between environmental variables and the lipids is the most novel part of this study. I suggest that the authors analyze univariate relationships of all possible variables.

Result and Discussion: The most interesting part of this study is the relationships between the fatty acids and environmental variables, but, as stated above, the authors didn’t analyze this data extensively, which has left this portion of the paper quite weak. The authors should perform additional univariate analyses between environmental variables and the fatty acid groupings. The relationships between environmental variables and fatty acids could also be invoked to be explain the weak correlations between the ELFA and PLFAs.

Response: We were concerned about the large p small n problem at first. But after consulting with statistical experts, we believe performing bivariate correlations for all environmental variables is acceptable under the context of this study. We have remade Fig. 4, added a justification for using all environmental variables on lines 163-167, and made the changes on lines 253-257 and 268-270. 

Results and Discussion:

Currently, I find the authors claims that high organic matter accounts for the lack of correspondence between the two methods to be unconvincing for several reasons. First, the SOC in their soils ranged from 2.1-3.8%. This is not unusually high and is within the range of other studies that have found good agreement between the two methods. Additionally, the two techniques were previously compared in forest soils in a study that found much stronger correlations than in the current study. Agreement was also found between the methods in compost, which was presumably very high in organic matter. The authors could support this argument if they showed that ELFA was more strongly correlated with SOC than was PLFA, but they have not performed this analysis.

Response: Thank you for your suggestion. We assume that the reviewer was talking about initial lines 218-220. We have revised lines 223-236. We focus on specific fatty acids and avoid mentioning their sources and implicitly linking SOC with the two methods, if any. Even though in the updated Fig. 4, SOC had a weak relationship with both ELFA-derived biomass (r = -0.12) and PLFA-derived biomass (r = -0.45), this finding cannot rule out the possibility that living microbes do not account for the majority of fatty acids detected by ELFA in this study. Since both methods only target a small proportion of SOC, SOC do not necessarily have a positive relationship with ELFA- or PLFA-extractable fatty acids. 

Figure 2a: “SOC” and “CN Ratio” are overlapping in this figure, making it difficult to read. Modify the figure so these words are not overlapping.

Response: We have updated this figure.

---

## [Editor Report · Decision Letter 1]

28 Apr 2021

Ester linked fatty acid (ELFA) method should be used with caution for interpretating soil microbial communities and their relationships with environmental variables in forest soils

PONE-D-20-33719R1

Dear Dr. Kang,

We’re pleased to inform you that your manuscript has been judged scientifically suitable for publication and will be formally accepted for publication once it meets all outstanding technical requirements.

Kind regards,

Daniel Cullen

Academic Editor

PLOS ONE
---

## [Editor Report · Acceptance letter]

30 Apr 2021

PONE-D-20-33719R1 

Ester linked fatty acid (ELFA) method should be used with caution for interpretating soil microbial communities and their relationships with environmental variables in forest soils 

Dear Dr. Kang:

I'm pleased to inform you that your manuscript has been deemed suitable for publication in PLOS ONE. Congratulations! Your manuscript is now with our production department. 

Kind regards, 

on behalf of

Dr. Daniel Cullen 

Academic Editor

PLOS ONE